# The Effects of Laser Remelting on the Microstructure and Performance of Bainitic Steel

**Yuelong Yu [1], Min Zhang [1,\*], Yingchun Guan [2,\*] , Peng Wu [3], Xiaoyu Chong [3], Yuhang Li [2] and Zhunli Tan [1]**

1   School of Mechanical, Electronic and Control Engineering, Materials Science & Engineering Research Center, Beijing Jiaotong University, Beijing 100044, China
2   School of Mechanical Engineering and Automation, Beihang University, 37 Xueyuan Road, Beijing 100191, China
3   Faculty of Material Science and Engineering, Kunming University of Science and Technology, Kunming 650093, China
\*   Correspondence: zhangm@bjtu.edu.cn (M.Z.); guanyingchun@buaa.edu.cn (Y.G.); Tel.: +86-10-5168-5462 (M.Z.); +86-10-8231-7430 (Y.G.)

**Abstract:** The surface of bainitic steel was remelted by fiber laser, and the microstructure and mechanical properties of the melted layer were studied by scanning electron microscopy (SEM), transmission electron microscopy (TEM), a nanoindentation instrument, and wear equipment. The study of changing the laser scanning speed showed that the depth of the melted layer increases with decreases of the laser scanning speed. The wear-resistance property increased by 55% compared with the matrix and decreased with the reduction of laser scanning speed within a certain range. In the study of changing the laser-scanning space, the thermal effect of laser melting in the back channel on the front channel was further validated. At the same time, it was found that the solidified layer surface of hardness alternating with softness can be obtained by appropriately expanding the scanning space, which is conducive to improving the wear-resistant properties of the steel surface, and properly improving the production efficiency of the laser remelting treatment.

**Keywords:** laser remelting; bainitic steel; melting layer; melting process; wear-resistant property

## 1. Introduction

Bainitic steel has been widely studied, developed, and applied because of its diverse structure and various matchings of good performance [1–4]. However, due to the tough working environment of bainitic steel, such as the lining plate, toothed plate, ram hammer, spade tooth, etc., the surfaces are often subjected to heavy loads. Thus, the serious surface wear of bainitic steel is the main cause of failure [5–8]. As wear occurs on the surface, proper treatment and improvement of the surface properties are the most direct and effective solutions to the wear of bainitic steel. By using laser surface remelting (LSR) technology, the surface modification layers with uniform structure, fine grain crystals, and no pores, cracks, and other defects can be obtained while the toughness and plasticity of the core part can also be maintained, thus the strength and wear resistance of the material surface can be improved [9–11]. Tang et al. [12] prepared a dual-gradient bainitic steel matrix composite with a hardness and impact toughness of 510 HV and 75 J/cm$^2$, respectively, by direct laser deposition. Guo et al. [13,14] improved the surface hardness of bainitic steel by different degrees through isothermal transformation at different temperatures after laser remelting. Xing et al. [15–17] achieved a significant improvement of the surface hardness of bainitic steel by studying the influence of different laser scanning speeds on the modified layer. However, it is difficult to process the workpiece as a whole

due to the limitation of laser spot size. Therefore, multiple-pass laser surface remelting (MPLSR) are required. The structure and properties of laser remelting-modified layers are affected by technological parameters such as laser scanning speed and scanning space [18–21]. At present, data on MPLSR are rare. In this study, an analysis was conducted on the results of microstructure, hardness, friction, and wear tests to study the influence of scanning speed and scanning space of an optical fiber laser on the surface remelting of bainitic steel, which provides a reference for MPLSR strengthening technology.

## 2. Materials and Methods

Bainitic steel was used as the substrate material, and its main chemical composition is shown in Table 1. The heat treatment process of the material is shown in Figure 1. Samples were wire cut to size $19 \times 12 \times 12$ mm$^3$. The three laser processing parameters are shown in Table 2. Before remelting, the samples were grinded by 240 #, 400 #, and 600 # sandpaper in turn, then polished with 2.5 W polishing paste, and finally cleaned by acetone. The remelting method is shown in Figure 2. Using a fiber laser (Super-drilling 600F, Shengxiong Laser, Dongguan, China), the spot size was 100 μm, and the laser wavelength was 1060–1100 nm. The processing path is shown in Figure 3a. In order to reduce the evaporation and oxidation of surface materials in the treatment process, argon was used to protect the laser molten pool. After mechanical polishing, the specimens were etched by a 4% nital solution for 15 s. The microstructure and wear surface were observed by optical microscopy (OM, Axio Vert.A1, Zeiss, Tokyo, Japan), scanning electron microscopy (SEM, EVO18, Zeiss, Tokyo, Japan), and transmission electron microscopy (TEM, JEM-F200, JEOL, Tokyo, Japan). The TEM cross-section sample was prepared by adhesion of two slices and then ion milling. Nanoindentation hardness of the surface and cross section of the modified specimens was measured by an iNano (iMicro, Oak Ridge, California, USA) nanoindentation instrument. The NanoBlitz three-dimensional (3D) release method implemented in the iNano instrument was used to perform an array of $20 \times 20$ indentations within a square area of $400 \times 400$ μm on the surface of this specimen. Indentations were performed to a peak force of 10 mN at a rate of approximately 1 indent per second [22]. In the wear test, the ring block wear tester (MRH-5A, Jinan, China) was used. The ring sample was GCr15, the hardness after heat treatment was 61 HRC, the block sample was bainitic steel, and the wear load was 100 N. Samples were weighed before and after wear with an accuracy of 0.01 mg using an electronic balance (sartorius).

**Table 1.** Chemical compositions of steel (wt%).

| Element | C | Mn | Si | Cr | Ni | Mo | Fe |
|---------|-----|-----|---------|---------|---------|---------|------|
| wt% | 0.2 | 2.0 | 0.5–1.5 | 0.5–1.0 | 0.4–0.8 | 0.3–0.6 | Bal. |

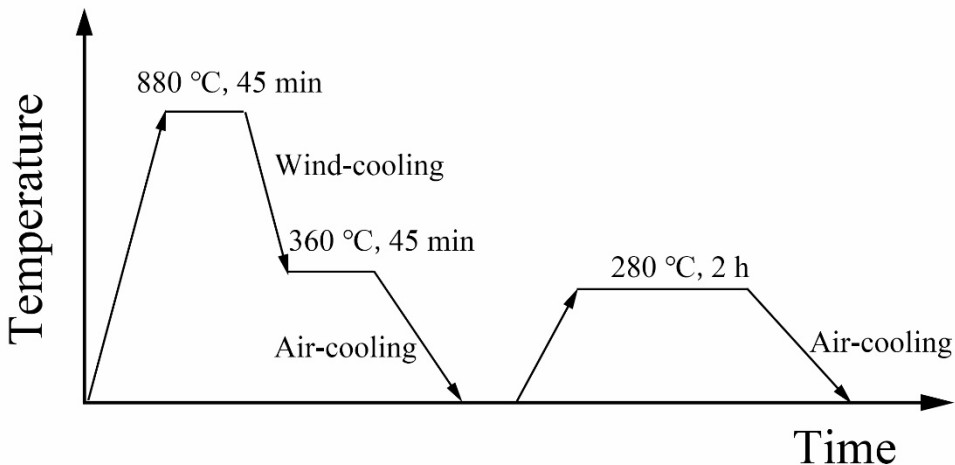

**Figure 1.** Heat treatment process.

| Sample Number | Laser Power (W) | Scanning Speed (mm/s) | Scanning Space (μm) | Shielding Gas |
|---|---|---|---|---|
| P-1# | 90 | 110 | 80 | Ar |
| P-2# | 90 | 80 | 80 | Ar |
| P-3# | 90 | 50 | 80/90/100/110/120/130/140/150 | Ar |

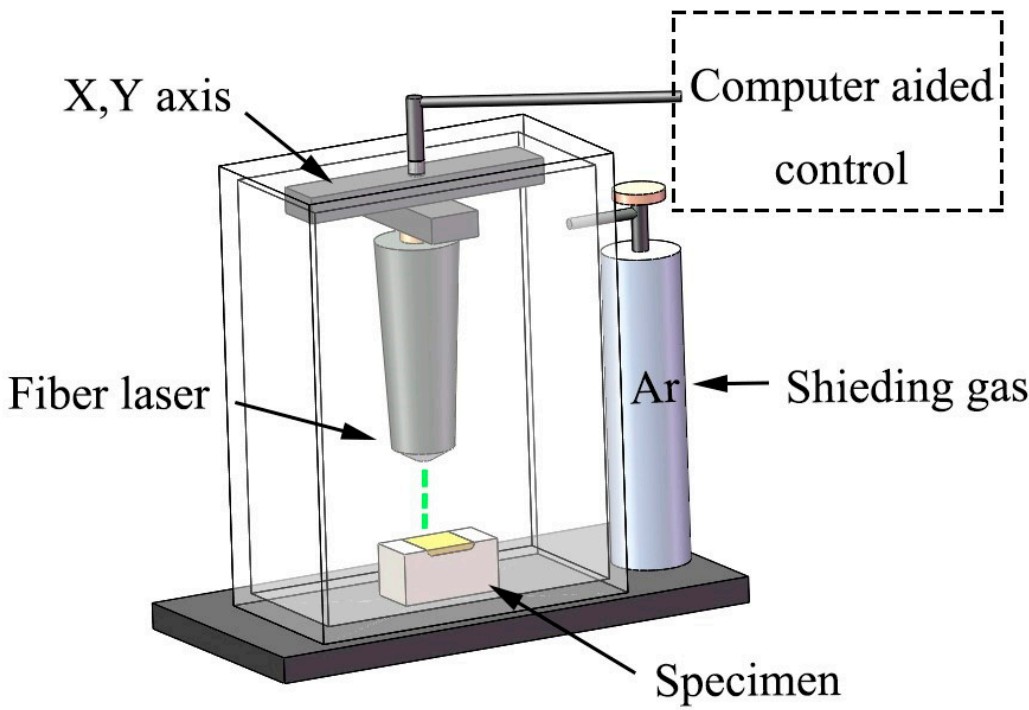

**Figure 2.** Equipment schematic of the LSR (laser surface remelting).

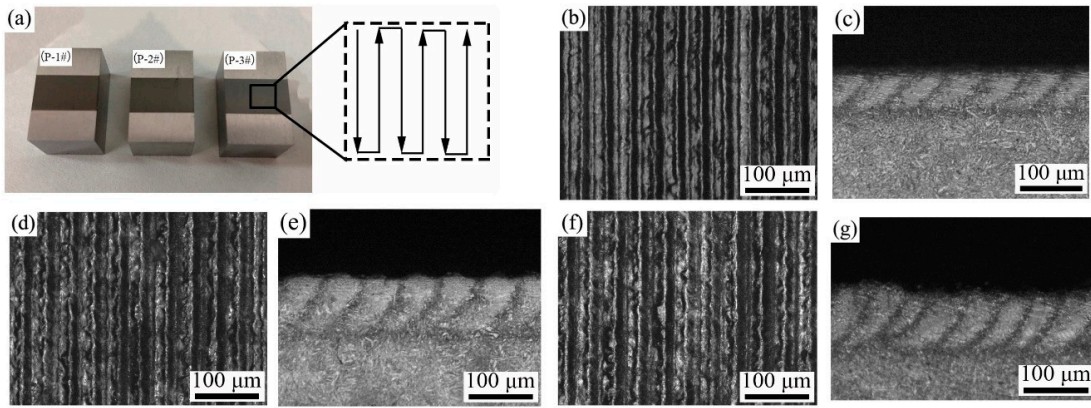

**Figure 3.** Morphology of bainitic steel after LSR. (**a**) Macroscopic overall morphology; (**b**–**g**) are the surface and cross-section micro-morphology of P-1 #, P-2 #, P-3 #, respectively.

## 3. Results and Discussion

### 3.1. LSR with Different Laser Scanning Speeds

The surface and section morphology of bainitic steel specimens irradiated with different laser scanning speeds are shown in Figure 3. It can be found that the recasting and microgroove stripes of molten materials are very obvious on the surface of the samples. As shown in Figure 3b–g, the surface and the cross section of the samples varies with the change of laser scanning speed, and the remelted

surface gradually becomes disorderly with high undulation. Figure 4 shows that the depth of the modified layer of samples increases with the decrease of scanning speed. This is because the heat input increases with the decrease of laser moving speed. The heat input or energy density (H) in the laser irradiation area can be estimated by using the following methods:

$$H = \frac{P}{2\omega_0 v} \tag{1}$$

where $P$, $\omega_0$ and $v$ represent laser power, spot radius, and scanning speed, respectively [23]. Therefore, as has been reported in the literature [24,25], high heating energy will not only introduce deep modified areas, but also evaporate some materials, such as Mn, Si, and Cr elements, especially in the center of the laser scanning path.

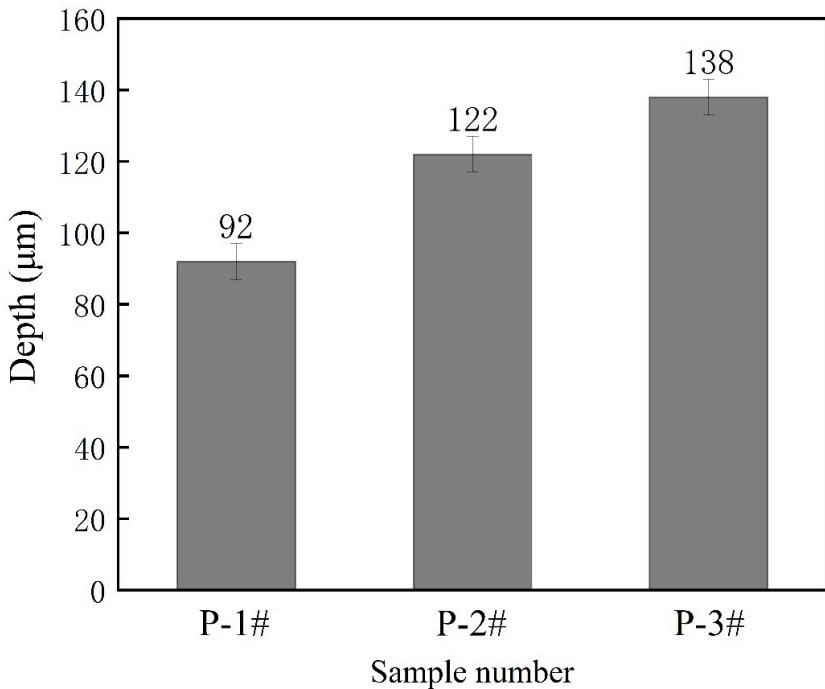

**Figure 4.** Depth of modified layer by LSR.

Taking the scanning speed of 50 mm/s as an example, the microstructures of the typical single-pass melted region were analyzed, as shown in Figure 5a. The energy distribution of the laser spot was uneven, with the center being high, the edge low, and the modified layer a crescent. Obviously, along the depth direction of the cross section, the laser modified layer can be divided into a complete transformation region and an incomplete transformation region; these are attributed to the fact that the temperature of melting zone is higher than Ac3 during laser processing, so the nucleation and growth of crystals occur in the matrix. The heat generated by laser melting transfers rapidly to the base metal and air, so rapid cooling is achieved. Specifically, in laser processing, the heating time and peak temperature of the surface are higher than the solid/liquid interface. Consequently, for the area near the surface, high energy intake leads to solid melting and then rapid cooling under the dual action of the air and the matrix, which is conducive to the formation of martensite. However, for the subsurface layer, the martensitic transformation in the cooling stage is not complete due to the relatively incomplete austenitization in the heating stage. For the laser melting process, the heating rate and peak temperature dynamics in the heating stage define the final microstructures. As shown in Figure 5b, the microstructure characteristics of martensite obtained by rapid heating and cooling can be observed, which is similar to the most widely accepted microstructure in bainite/martensite duplex steels. Figure 5c illustrates the transition form from uniform lath martensite without strip-like

characteristics to acicular martensite. Figure 5d shows the structure of the heat affected zone (HAZ), where larger acicular martensite is obtained due to the higher temperature and longer residence time at the high temperature [26,27]. Figure 5e shows a matrix structure consisting mainly of lath bainite (LB) and a few martensite/austenite (M/A) islands [2,3]. The micrographs of the typical TEM of different regions of single-pass melting are shown in Figure 6. The ultra-thin martensite lath structure produced by rapid cooling at a distance of 50 μm from the surface is the main reason for the sharp increase of hardness. With the change of depth, the peak temperature of different irradiated regions is different, resulting in the obvious change of the substructure of martensite [28–30]. It can be found in Figure 6c that the typical fine structure is dislocation. Whereas in Figure 6d, twinned martensite is observed. This is a typical selected area electron diffraction pattern of a {112} <111> twin with beam direction parallel to a [−113] α-martensite twin structure. A cross-section hardness gradient after single-pass LSR is plotted in Figure 7. It can be seen that the hardness of the modified layer is obviously higher than that of the matrix, and decreases as a whole from the surface to the inside. The maximum hardness value of 6.8 GPa appears at 50 μm away from the surface, where the microstructure is a fine martensite lath structure, and the change trend of hardness is consistent with the analysis results of the microstructure.

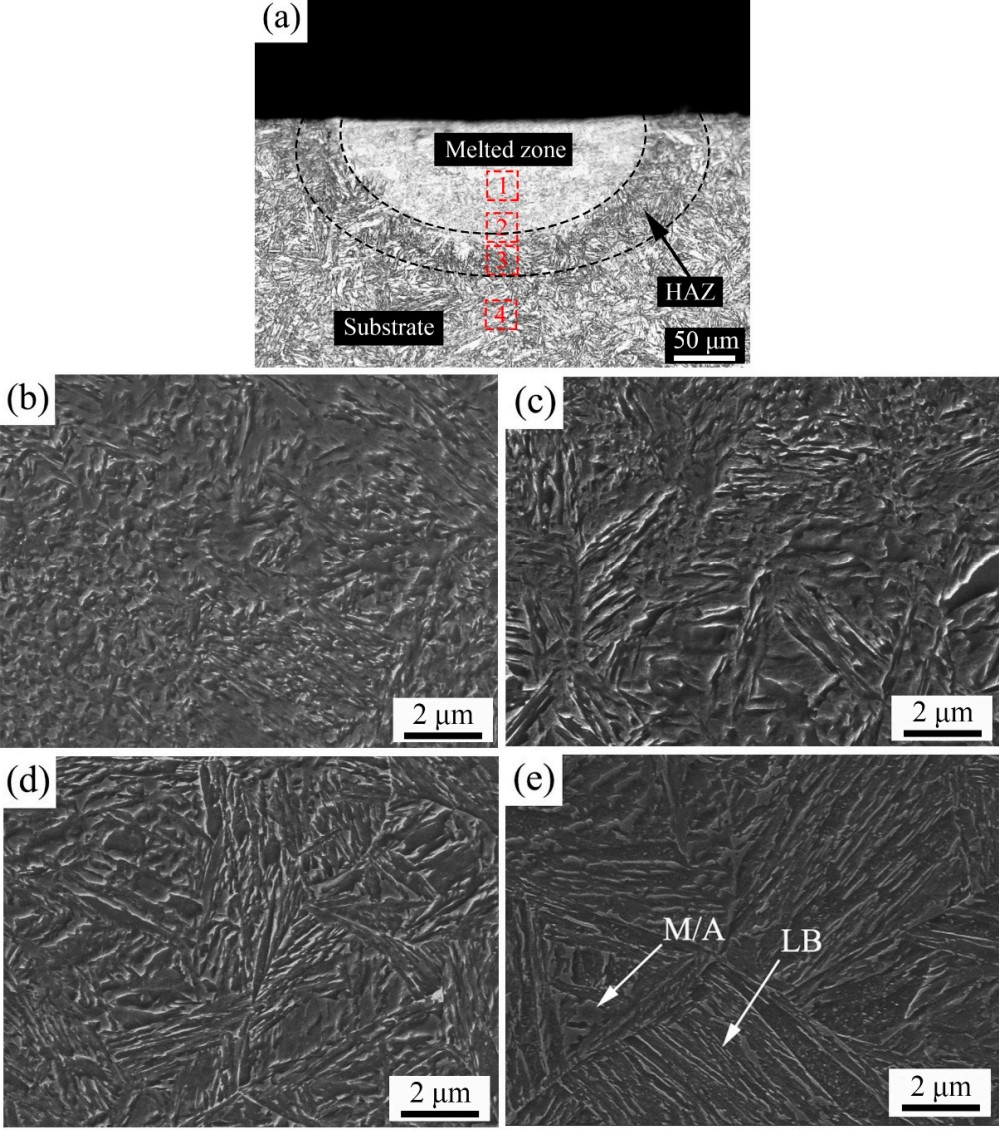

**Figure 5.** Microstructure and morphology of the laser single-pass modified layer. (**a**) cross-sectional panorama; (**b**–**e**) group morphology corresponding to the 1–4 positions in (**a**).

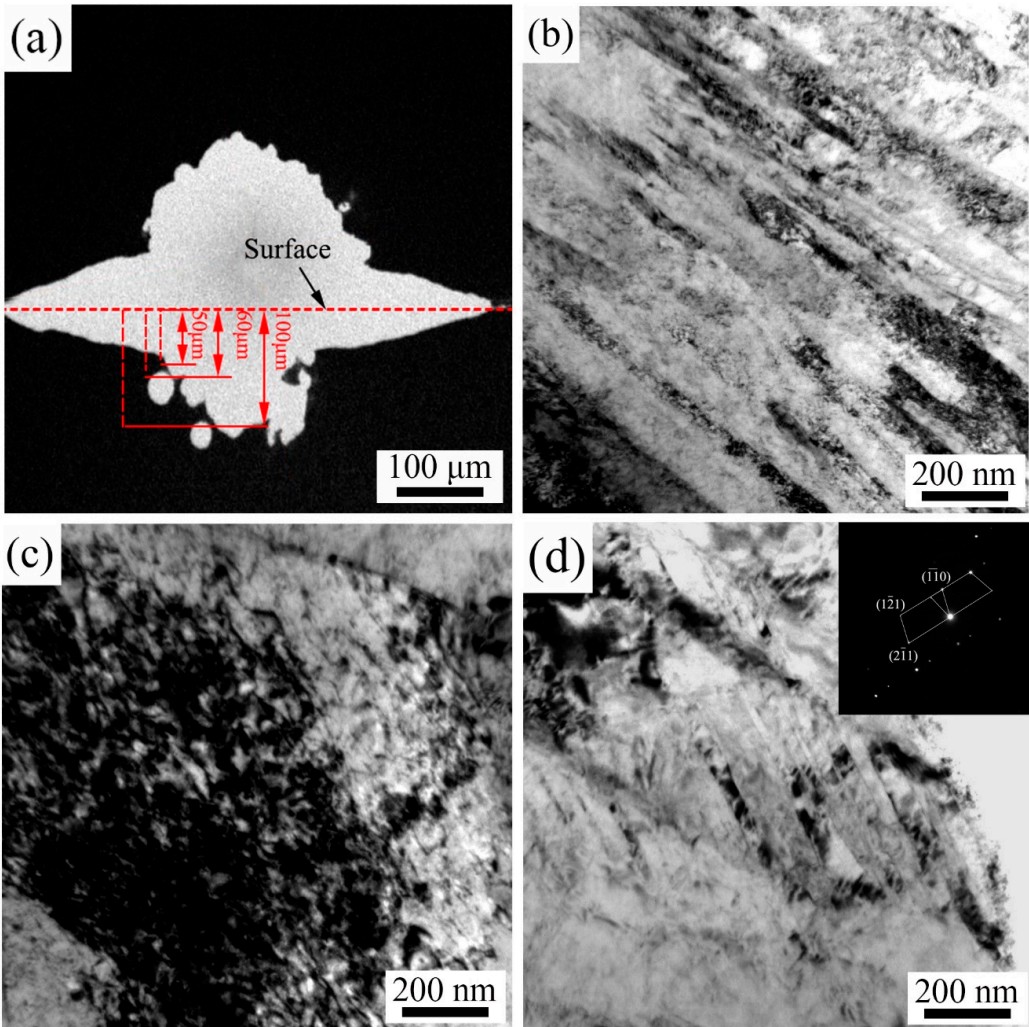

**Figure 6.** TEM (transmission electron microscopy) images of different areas. (**a**) The low-magnification TEM micrograph of the cross-section specimen. It is used to label the distance from the surface. The distance between (**b**) and the surface is 50 μm, between (**c**) and the surface is 60 μm, and between (**d**) and the surface is 100 μm.

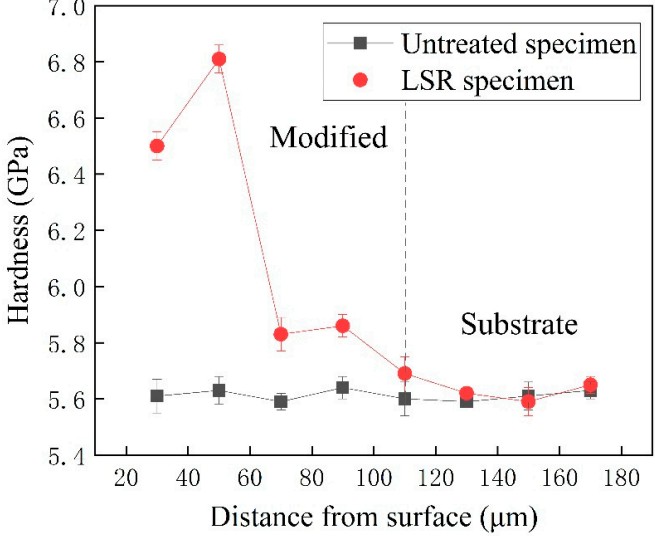

**Figure 7.** Hardness gradient of bainitic steel after LSR.

Figure 8a shows the wear pattern of the specimen. The samples before and after the wear test were weighed and the results are shown in Figure 8b. The wear weight loss and wear trace depth of the modified layer after LSR were lower than those of the matrix. When the scanning speed was 110 mm/s, the wear resistance of the bainitic steel surface after LSR was enhanced to the maximum, the wear trace depth was reduced by 47%, and wear weight loss was reduced by 55% compared with that of the untreated sample. Therefore, the LSR treatment of bainitic steel can significantly improve its surface wear resistance, and within a certain scanning speed range, the larger the scanning speed, the stronger the wear resistance of the melted surface. The wear surface morphology of the four groups of specimens was observed by SEM. It can be found that the wear surfaces of the four groups of specimens all present ditches and peeling. From Figure 9a, a large amount of debris is accumulated on the wear surface of the sample without LSR, and this debris peeling will greatly reduce the wear resistance of the surface. This is visualized in Figure 9c,d, in which the LSR can effectively improve the wear resistance of the specimens under the same sliding wear condition. With the increase of laser scanning speed, debris peeling decreases gradually. For the specimen with MPLSR in the friction process, the micro-convex body in the hardened layer is subjected to continuous cyclic shear stress, which results in fracture and the abjunction of the micro-convex body, leading to gradual formation of adhesive wear. However, in a certain range, with the increase of laser scanning speed, the transition of hard and soft microstructures in the modified layer gradually becomes continuous, which increases the wear resistance [7].

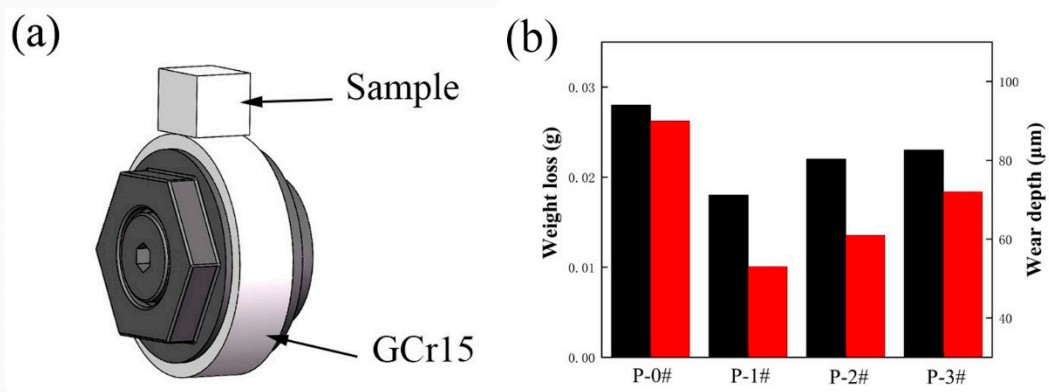

**Figure 8.** Wear of the modified layer with different laser scanning speeds. (**a**) Illustration of the wear contact mode of the ring-block; (**b**) Weight loss and wear depth after LSR.

## 3.2. LSR with Different Scanning Space

Scanning space is the main parameter affecting the performance and productivity of MPLSR. The influence of different scanning spaces on the melting layer was further studied. In order to determine the principle of selecting the scanning space, first, a simple model as shown in Figure 10 was established according to the macro-morphology of the cross section of the MPLSR specimen with a scanning speed of 50 mm/s. Figure 10 is a schematic diagram of the lap joint of melting by two lasers in which the solid line is the boundary of the molten pool, the dotted line is the boundary of the HAZ, and the shadow is the lap area. Two characteristic parameters were set, in which 'd' is the scanning space of two meltings and 'e' is the width of the HAZ.

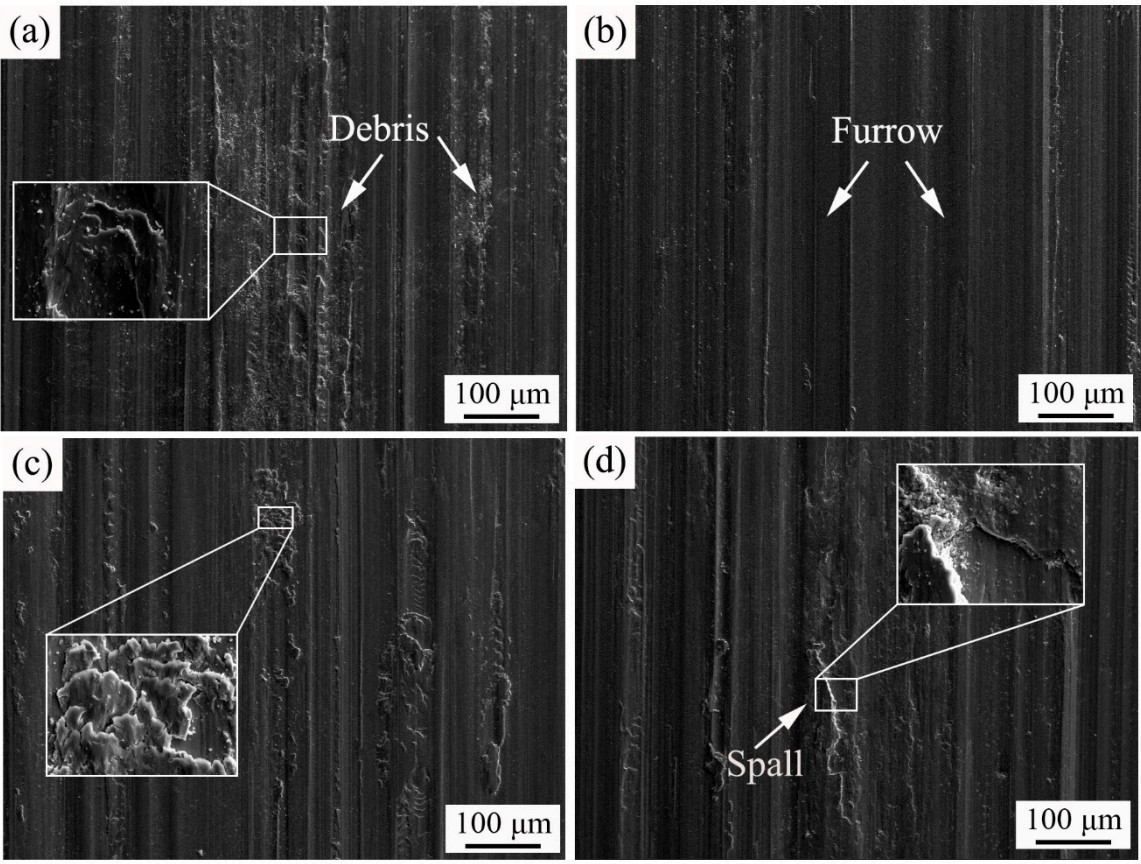

**Figure 9.** SEM morphology of the surface wear of specimens. (**a**) sample P-0#, (**b**) sample P-1#, (**c**) sample P-2#, (**d**) sample P-3#.

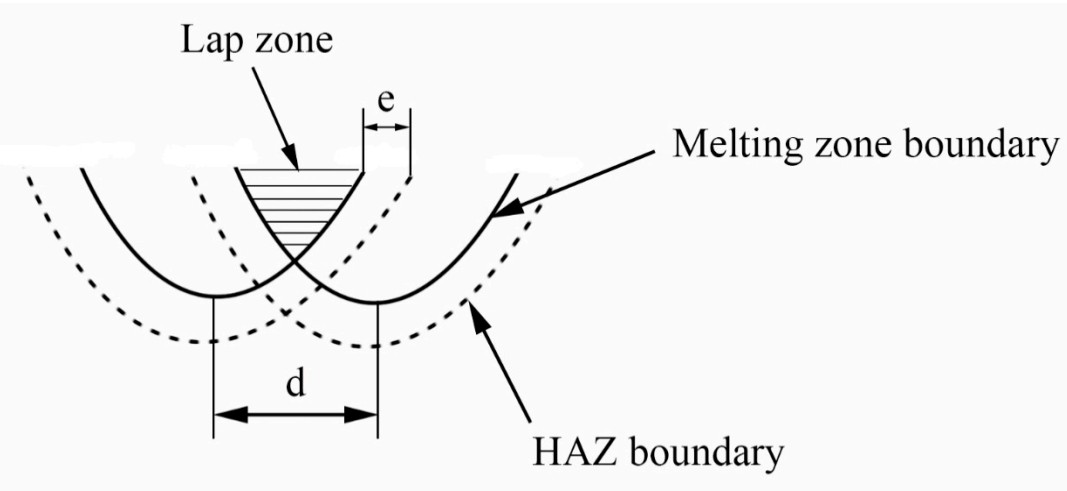

**Figure 10.** Simplified model of the multi-pass laser remelting.

The lap zone is the overlap zone between the second and the first melting zones during LSR treatment, as shown in the shadow of Figure 10. Due to the effect of the second laser heating, the lap zone has a tempering-softening effect. The softening band caused by the lap may affect the wear resistance and service life of the material. The nanoindentation hardness test at 20 μm below the surface of the melting layer was conducted, and the indentation array results are shown in Figure 11. It can be seen that the varied range of the hardness of the overlap zone does not exceed 0.8 GPa, and the hardness of the MPLSR presents a high distribution alternating with a low distribution, which

indicates that there is an obvious tempering-softening effect on the front pass from the back pass in the MPLSR treatment. From Figure 11, it can be seen that the hardness of the lap zone and laser melting zone is obviously higher than that of the parent metal, which suggests that the tempering-softening in lap zone is not serious. During melting, the second heating promotes part of the transformed first hardening structure to produce tempering-softening. In addition, because of the Gauss distribution of the laser spot energy, the temperature of the laser beam center decreases gradually from inside to outside, and the uneven heating of the overlap part inevitably results in a series of tempering transformations of different degrees, thus different hardness distributions are formed. As shown in Figure 12, it can be seen from the analysis of the wear test results that the wear resistance of the LSR specimens is improved with the increase of scanning space within a certain range. In the wear test, the friction test was performed on the surface of the melting sample in the form of a line contact and the frictional resistance was mainly borne by the hardened zone with high hardness. The higher the surface hardness is, the less abrasive it is, and better wear resistance it has. The softening zone connected with the hardening zone has the function of cushioning and unloading the force, which is conducive to improving the overall wear resistance. With the further increase of scanning space, the inhomogeneity of the microstructure and hardness of the modified layer also increase, which leads to worse wear resistance. Consequently, properly increasing the scanning space is conducive to improving wear resistance and production efficiency. Under the experimental conditions in this study, the suitable scanning space was 130 µm.

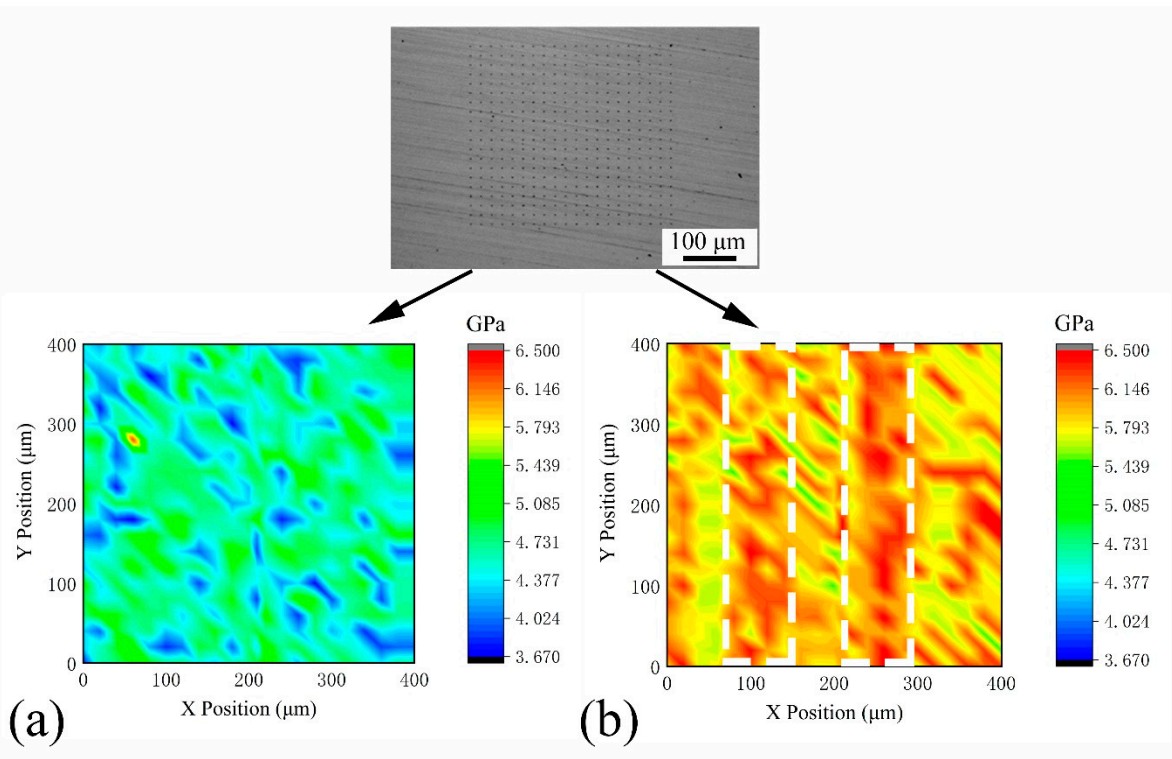

**Figure 11.** Distribution of the hardness values of the bainitic steel; (**a**) untreated specimen and (**b**) LSR specimen with $d = 100$ µm.

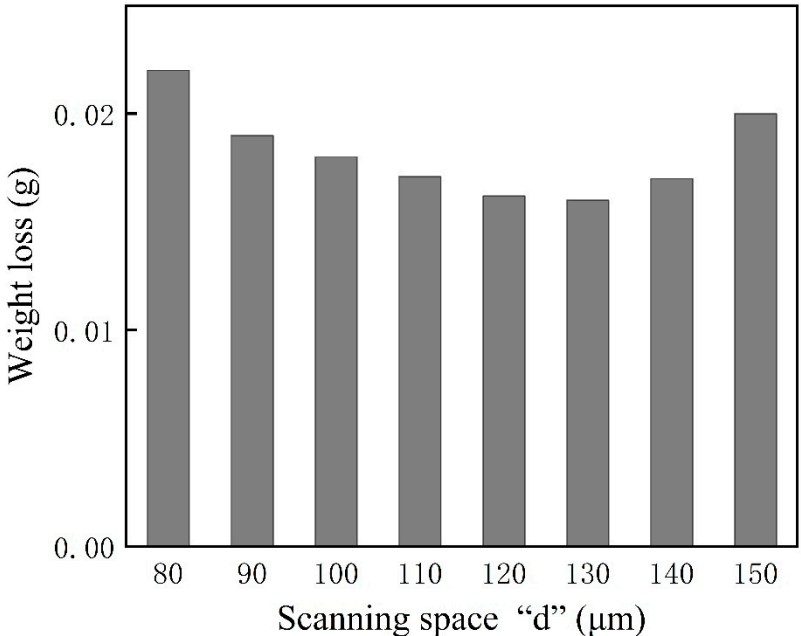

**Figure 12.** Weight loss with different scanning spaces.

## 4. Conclusions

In summary, through the LSR treatment of bainitic steel and the analysis of the tests of the modified layer, it was found that in the experiments with different laser scanning speeds and the same scanning space, the wear resistance can be increased by a maximum of 55% compared with the matrix, and decreases gradually with the decrease of the scanning speed. When the scanning speed was 50 mm/s, the maximum hardness increased by 1.2 GPa compared with the matrix. In the experiments with the same scanning speed and different scanning space, the hardness distribution showed the heat effect on the front pass of the back pass melting. At the same time, the modified layer surface with hardness alternating with softness can be obtained by a proper increase of the scanning space, which is conducive to improving the wear resistance and productivity.

**Author Contributions:** Conceptualization, Y.Y. and M.Z.; methodology, Y.Y. and M.Z.; validation, Y.L.; investigation, Y.Y.; resources, M.Z.; data curation, Y.Y. and M.Z.; writing-original draft preparation, Y.Y.; writing-review and editing, M.Z., Y.G. and P.W.; supervision, Y.G., X.C. and Z.T.

**Funding:** The research was supported by the Fundamental Research Funds for the Central Universities, grant number 2019JBM044 and the Fund of Key Laboratory of Advanced Materials of Ministry of Education, grant number XJCL201908.

**Conflicts of Interest:** The authors declare no conflict of interest.

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
