# Peer review of "The Effects of Laser Remelting on the Microstructure and Performance of Bainitic Steel"

_metals, doi:10.3390/met9080912_

Round 1

Reviewer 1 Report

P4 L91

The characters in the equation (1) are not listed.

P4 L92

What elements are evaporated? And reviewer want an explanation of what happens when it evaporates.

P5 L107

There is a description that "the heating time and peak temperature of the area near the interface are higher than then below the surface.", but where are the “interface” and the “surface”? If “interface” is the interface between the melt and the matrix, and “surface” points to the surface of the melt, the phenomenon that the temperature at “interface” is higher than “surface” can not be understood.

P5 L113 to L115

What is the basis that the author described "fine lath martensite"? What can be observed by normal OM and SEM observation is the prior austenite grains, packets and blocks.

P5 L116

It differs from the general HAZ appearance. The explanation that the coarse particles are caused by quenching is not accepted.

P6 Figure 5(a)

Scale bar should be added.

P5 L118

The author described that “Figure 5e shows ... few martensite/austenite (M/A) islands.”, but how was it judged that it is austenite?

P6 L124

Generally, when martensite changes from lath to twin(lens), the amount of carbon, not the heat treatment conditions, will affect. Since, normally, the lath martensite is formed in the steel with 0.2wt% carbon, it is necessary to provide a supplementary explanation about the TEM observation results, if the author insist that twins are generated.

P7 Figure 6(a)

How was the TEM thin film prepared? And, according to other data, the size of the single-pass melting area seems to be about 100 μm in radius. However, the hole in the TEM sample seems to be of similar size to it. So, did you confirm that the observation field of view “50 μm from the surface” is in the melting area?

P8 L156

Does this explanation contradict the conclusion in Section 3.2 that "the alternate presence of soft and hard improves the wear resistance"?

P9 Figure 9

There is no description about the sample P-0#.

P10 L174

The author states that "when d = 100 μm, the boundary of the melting zones intersects exactly", but as far as Figure 11 is concerned, it overlaps significantly.

P10 L175

The author states that "when d = 150 μm, the boundary of the HAZ intersects.", But where is the HAZ in Figure 11(c)? Please show in Figure 11.

P10 L182

Since the "lap zone" is melted by the second LSR, it should be understood that the microstructure of that part is fresh martensite.

P11 L201

Since the hardness distribution of the surface is shown only for d = 100 μm, the explanation on this line has no basis.

Reviewer 2 Report

Abstract: Re-write the first sentence.
Line 20: "The wear-resisting property increases by 55% compared with the maximum of matrix". What does it mean?
Line 22: Re-write
Line 54: Bainitic steel is used as substrate material
Line 76: Chemical compositions of steel (wt%)
Line 55: Samples are wire cut...
Line 56: mm3 (use superscript for 3)
Line 56: "The surface is cleaned and polished by acetone before remelting". Are samples polished only by Acetone? if sandpaper or any other polishing material is used please specify. Samples are cleaned after polishing or before?
Line 61: etched instead of corroded.

Author Response

Reply to Reviewer’s Comments

Metals

Manuscript: Metals-574150
Title: Effect of laser remelting on microstructure and performance of bainitic Steel

Authors: Yuelong Yu, Min Zhang, Yingchun Guan, Peng Wu,

Xiaoyu Chong ,Yuhang Li, Zhunli Tan

We are very grateful to the reviewers for the valuable comments. We have carefully considered the comments and made modifications in the manuscript accordingly as listed below. All the modifications which have been done are highlighted with in BLUE colour in the revised manuscript.

Note: In the authors’ reply to the reviewers, the paper numbers, figure numbers, and references relate to the revised manuscript.

Reviewers' comments:

Reviewer #2:

Abstract: Re-write the first sentence.

Reply: According to the reviewer’s suggestion, we have revised this sentence to “The surface of bainitic steel was remelted by fiber laser, in which the microstructure and mechanical properties of the melted layer were studied by scanning electron microscopy (SEM), transmission electron microscopy (TEM), nanoindentation instrument and wear equipment”.

Line 20: "The wear-resisting property increases by 55% compared with the maximum of matrix". What does it mean? 

Reply: we have revised this sentence to “The wear-resisting property increases by 55% to the greatest extent compared with the matrix and decreases with the reduction of laser scanning speed within a certain range”.

Line 22: Re-write

Reply: According to the reviewer’s suggestion, we have revised this sentence to “In the study of changing laser scanning space, the thermal effect of laser melting in the back channel on the front channel is further validated”.

Line 54: Bainitic steel is used as substrate material

Reply: corrected

Line 76: Chemical compositions of steel (wt%)

Reply: corrected

Line 55: Samples are wire cut...

Reply: corrected

Line 56: mm3 (use superscript for 3)

Reply: corrected

Line 56: "The surface is cleaned and polished by acetone before remelting". Are samples polished only by Acetone? if sandpaper or any other polishing material is used please specify. Samples are cleaned after polishing or before?

Reply: Thanks for the reviewer’s constructive suggestion. we have revised this sentence to “Before remelting, the samples were grinded by 240#, 400#, 600# sandpaper in turn, then polished with 2.5W polishing paste, and finally cleaned by acetone”.

Line 61: etched instead of corroded.

Reply: corrected

Round 2

Reviewer 1 Report

The author's sincere and polite answers and revises can be fully appreciated. Only one word needs to be corrected.

P8 L159 "tissues" should be modified to "microstructures".

Author Response

According to the reviewer’s suggestion, we have revised this word to "microstructures"

thank you so much

Reviewer 2 Report

Line 90-Eq (1): Please demonstrate how the H is derived. My calculation shows that the H=4P/(πω0V).

Figure 4: The Y axis title needs to be Depth (μm) instead of "Depth/μm".

Line 111: "Consequently, for the area near the surface, the austenitizing process is more complete, which is conducive to martensitic transformation". What does it mean? and Why this is happening?

Figure 7: The X axis title needs to be Distance from surface (μm) instead of "Distance from surface/μm". Please apply that to all relevant figures and tables.

Author Response

See PDF for details. Thank you 
